# TEACH ME HOW TO INTERPOLATE A MYRIAD OF EMBEDDINGS

## ABSTRACT

*Mixup* refers to interpolation-based data augmentation, originally motivated as a way to go beyond *empirical risk minimization* (ERM). Yet, its extensions focus on the definition of interpolation and the space where it takes place, while the augmentation itself is less studied: For a mini-batch of size $m$, most methods interpolate between $m$ pairs with a single scalar interpolation factor $\lambda$.

In this work, we make progress in this direction by introducing *MultiMix*, which interpolates an arbitrary number $n$ of tuples, each of length $m$, with one vector $\lambda$ per tuple. On sequence data, we further extend to *dense* interpolation and loss computation over all spatial positions. Overall, we increase the number of tuples per mini-batch by orders of magnitude at little additional cost. This is possible by interpolating at the very last layer before the classifier. Finally, to address inconsistencies due to linear target interpolation, we introduce a *self-distillation* approach to generate and interpolate synthetic targets.

We empirically show that our contributions result in significant improvement over state-of-the-art mixup methods on four benchmarks. By analyzing the embedding space, we observe that the classes are more tightly clustered and uniformly spread over the embedding space, thereby explaining the improved behavior.

## 1 INTRODUCTION

*Mixup* (Zhang et al., 2018) is a data augmentation method that interpolates between pairs of training examples, thus regularizing a neural network to favor linear behavior in-between examples. Besides improving generalization, it has important properties such as reducing overconfident predictions and increasing the robustness to adversarial examples. Several follow-up works have studied interpolation in the *latent* or *embedding* space, which is equivalent to interpolating along a manifold in the input space (Verma et al., 2019), and a number of nonlinear and attention-based interpolation mechanisms (Yun et al., 2019; Kim et al., 2020; 2021; Uddin et al., 2021; Hong et al., 2021). However, little progress has been made in the augmentation process itself, *i.e.*, the number of examples being interpolated and the number of interpolated examples being generated.

Mixup was originally motivated as a way to go beyond *empirical risk minimization* (ERM) (Vapnik, 1999) through a vicinal distribution expressed as an expectation over an interpolation factor $\lambda$, which is equivalent to the set of linear segments between all pairs of training inputs and targets. In practice however, in every training iteration, a single scalar $\lambda$ is drawn and the number of interpolated pairs is limited to the size of the mini-batch, as illustrated in Figure 1(a). This is because, if interpolation takes place in the input space, it would be expensive to increase the number of examples per iteration. To our knowledge, these limitations exist in all mixup methods.

In this work, we argue that a data augmentation process should augment the data seen by the model, or at least by its last few layers, as much as possible. In this sense, we follow *manifold mixup* (Verma et al., 2019) and generalize it in a number of ways to introduce *MultiMix*, as illustrated in Figure 1(b). First, rather than pairs, we interpolate *tuples* that are as large as the mini-batch. Effectively, instead of linear segments between pairs of examples in the mini-batch, we sample on their entire *convex hull*. Second, we draw a different vector $\lambda$ for each tuple. Third, and most important, we increase the number of interpolated tuples per iteration by orders of magnitude by only slightly decreasing the actual training throughput in examples per second. This is possible by interpolating at the deepest

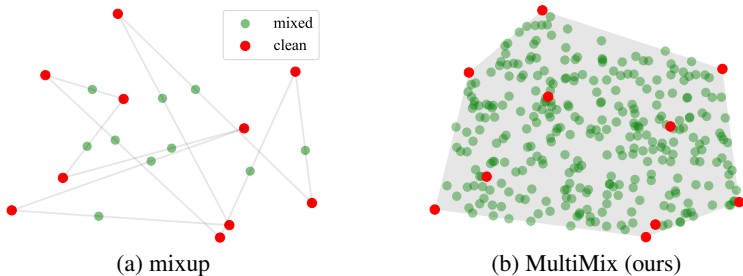

Figure 1: Data augmentation from a mini-batch $B$ consisting of $m = 10$ points in two dimensions. (a) *mixup*: sampling of $m$ points on linear segments between $m$ pairs of points in $B$, using the same interpolation factor $\lambda$. (b) *MultiMix*: sampling of $n = 300$ points in the convex hull of $B$.

layer possible, *i.e.*, just before the classifier, which also happens to be the most effective choice. The interpolated embeddings are thus only processed by a single layer.

Apart from increasing the number of examples seen by the model, another idea is to increase the number of loss terms per example. In many modalities of interest, the input is a *sequence* in one or more dimensions: pixels or patches in images, voxels in video, points or triangles in high-dimensional surfaces, to name a few. The structure of input data is expressed in matrices or tensors, which often preserve a certain spatial resolution until the deepest network layer before they collapse *e.g.* by global average pooling (Szegedy et al., 2015; He et al., 2016) or by taking the output of a classification token (Vaswani et al., 2017; Dosovitskiy et al., 2020).

In this sense, we choose to operate at the level of *sequence elements* rather than representing examples by a single vector. We introduce *dense MultiMix*, which is the first approach of this kind in mixup-based data augmentation. In particular, we interpolate densely the embeddings and targets of sequence elements and we also apply the loss densely, as illustrated in Figure 2. This is an extreme form of augmentation where the number of interpolated tuples and loss terms increases further by one or two orders of magnitude, but at little cost.

Finally, linear interpolation of targets, which is the norm in most mixup variants, has a limitation: Given two examples with different class labels, the interpolated example may actually lie in a region associated with a third class in the feature space, which is identified as *manifold intrusion* (Guo et al., 2019). In the absence of any data other than the mini-batch, a straightforward way to address this limitation is to devise targets originating in the network itself. This naturally leads to *self-distillation*, whereby a moving average of the network acts as a teacher and provides synthetic soft targets (Tarvainen & Valpola, 2017), to be interpolated exactly like the original hard targets.

In summary, we make the following contributions:

1. We introduce *MultiMix*, which, given a mini-batch of size $m$, interpolates an arbitrary number $n \gg m$ of tuples, each of length $m$, with one interpolation vector $\lambda$ per tuple—compared with $m$ pairs, all with the same scalar $\lambda$ for most mixup methods (subsection 3.2).
2. We extend to *dense* interpolation and loss computation over all spatial positions (subsection 3.4).
3. We use *online self-distillation* to generate and interpolate soft targets for mixup—compared with linear target interpolation for most mixup methods (subsection 3.3).
4. We improve over state-of-the-art mixup methods on *image classification*, *robustness to adversarial attacks*, *object detection* and *out-of-distribution detection* (section 4).

## 2  RELATED WORK

**Mixup: interpolation methods**   In general, mixup interpolates between pairs of input examples (Zhang et al., 2018) or embeddings (Verma et al., 2019) and their corresponding target labels. Several follow-up methods mix input images according to spatial position, either at random rectangles (Yun et al., 2019) or based on attention (Uddin et al., 2021; Kim et al., 2020; 2021), in an attempt to focus on a different object in each image. We also use attention in our dense MultiMix variant, but

in the embedding space. Other definitions of interpolation include the combination of content and style from two images (Hong et al., 2021) and the spatial alignment of dense features (Venkataramanan et al., 2022). Our dense MultiMix variant also uses dense features but without aligning them, hence it can mix a very large number of images and generate a large number of interpolated samples. Our work is orthogonal to these methods as we focus on the sampling process of augmentation rather than on the definition of interpolation.

**Mixup: sampling**   To the best of our knowledge, the only methods that interpolate more than two examples for image classification are OptTransMix (Zhu et al., 2020), SuperMix (Dabouei et al., 2021) and $\zeta$-Mixup (Abhishek et al., 2022). All three methods operate in the *input space* and limit the number of interpolated examples to the mini-batch size, $m$; whereas our MultiMix generates an arbitrary number of interpolated examples ($n = 1000$ in practice) in the *embedding space*. To determine the interpolation weights, OptTransMix involves a complex optimization process and only applies to images with clean background; $\zeta$-Mixup uses random permutations of a fixed vector; and SuperMix uses a Dirichlet distribution over *not more than* 3 samples in practice. We also sample weights from the Dirichlet distribution but interpolate as many examples as the mini-batch, $m$.

Beyond classification, $m$-Mix (Zhang et al., 2022) uses graph neural networks in a self-supervised setting with pair-based loss functions. The interpolation weights are deterministic and based on pairwise similarities. Essentially, this operation resembles a layer of a graph neural network.

**Self-distillation**   Distillation refers to a two-stage knowledge transfer process where a larger teacher model or ensemble is trained before predicting soft targets to train a smaller student model on the same (Buciluǎ et al., 2006; Hinton et al., 2015; Romero et al., 2014; Shen et al., 2019; Zagoruyko & Komodakis, 2016a) or different (Radosavovic et al., 2018; Xie et al., 2020) training data. The architecture of the two models may be the same with training at multiple stages, for example in continual learning (Rebuffi et al., 2017; Li & Hoiem, 2018). In self-distillation or co-distillation, not only the models are the same, but the knowledge transfer process is also online, *e.g.* between layers of the same model (Zhang et al., 2019) or between two versions of the model (Anil et al., 2018), where the teacher parameters may be obtained from the student rather than learned (Tarvainen & Valpola, 2017). The latter approach has been successful in self-supervised representation learning (Grill et al., 2020; Caron et al., 2021; Zhou et al., 2022).

As far as we know, distillation has only been used for mixup in SuperMix (Dabouei et al., 2021) as a two-stage process from a large pre-trained teacher model to a small student model. We are the first to use single-stage, online self-distillation in this context, following (Tarvainen & Valpola, 2017), where the teacher model is a moving average of the student.

**Dense loss functions**   Although standard in dense tasks like semantic segmentation (Noh et al., 2015; He et al., 2017), where dense targets commonly exist, dense loss functions are less common otherwise. Few examples are in few-shot learning (Lifchitz et al., 2019; Li et al., 2019), where data augmentation is of utter importance, and in unsupervised representation learning, *e.g.* dense contrastive learning (O Pinheiro et al., 2020; Wang et al., 2021), learning from spatial correspondences (Xiong et al., 2021; Xie et al., 2021a) and masked language or image modeling (Devlin et al., 2019; Xie et al., 2021b; Li et al., 2021; Zhou et al., 2022). Some of these methods use dense distillation (Xiong et al., 2021; Zhou et al., 2022), which is also studied in continual learning (Dhar et al., 2019; Douillard et al., 2020). To our knowledge, we are the first to use dense interpolation and a dense loss function for mixup. Our setting is supervised, similar to dense classification (Lifchitz et al., 2019), but we also use dense distillation (Zhou et al., 2022).

## 3   METHOD

### 3.1   PRELIMINARIES AND BACKGROUND

**Problem formulation**   Let $x \in \mathcal{X}$ be an input example and $y \in \mathcal{Y}$ its one-hot encoded target, where $\mathcal{X} = \mathbb{R}^D$ is the input space, $\mathcal{Y} = \{0, 1\}^c$ and $c$ is the total number of classes. Let $f_\theta : \mathcal{X} \to \mathbb{R}^d$ be an encoder that maps the input $x$ to an embedding $z = f_\theta(x)$, where $d$ is the dimension of the embedding. A classifier $g_W : \mathbb{R}^d \to \Delta^{c-1}$ maps $z$ to a vector $p = g_W(z)$ of predicted probabilities

over classes, where $\Delta^n \subset \mathbb{R}^{n+1}$ is the unit $n$-simplex, *i.e.*, $p \geq 0$ and $\mathbf{1}_c^\top p = 1$, and $\mathbf{1}_c \in \mathbb{R}^c$ is an all-ones vector. The overall network mapping is $f := g_W \circ f_\theta$.

Parameters $(\theta, W)$ are learned by optimizing over mini-batches. Given a mini-batch of $m$ examples, let $X = (x_1, \ldots, x_m) \in \mathbb{R}^{D \times m}$ be the inputs, $Y = (y_1, \ldots, y_m) \in \mathbb{R}^{c \times m}$ the targets and $P = (p_1, \ldots, p_m) \in \mathbb{R}^{c \times m}$ the predicted probabilities of the mini-batch, where $P = f(X) := (f(x_1), \ldots, f(x_m))$. The objective is to minimize the cross-entropy

$$H(Y, P) := -\mathbf{1}_c^\top (Y \odot \log(P)) \mathbf{1}_m / m \tag{1}$$

of predicted probabilities $P$ relative to targets $Y$ averaged over the mini-batch, where $\odot$ is the Hadamard (element-wise) product. In summary, the mini-batch loss is

$$L(X, Y; \theta, W) := H(Y, g_W(f_\theta(X))). \tag{2}$$

**Mixup** Mixup methods commonly interpolate pairs of inputs or embeddings and the corresponding targets at the mini-batch level while training. Given a mini-batch of $m$ examples with inputs $X$ and targets $Y$, let $Z = (z_1, \ldots, z_m) \in \mathbb{R}^{d \times m}$ be the embeddings of the mini-batch, where $Z = f_\theta(X)$. *Manifold mixup* (Verma et al., 2019) interpolates the embeddings and targets by forming a convex combination of the pairs with interpolation factor $\lambda \in [0, 1]$:

$$\widetilde{Z} = Z(\lambda I + (1 - \lambda)\Pi) \tag{3}$$

$$\widetilde{Y} = Y(\lambda I + (1 - \lambda)\Pi), \tag{4}$$

where $\lambda \sim \mathrm{Beta}(\alpha, \alpha)$, $I$ is the identity matrix and $\Pi \in \mathbb{R}^{m \times m}$ is a permutation matrix. *Input mixup* (Zhang et al., 2018) interpolates inputs rather than embeddings:

$$\widetilde{X} = X(\lambda I + (1 - \lambda)\Pi). \tag{5}$$

Whatever the interpolation method and the space where it is performed, the interpolated data, *e.g.* $\widetilde{X}$ (Zhang et al., 2018) or $\widetilde{Z}$ (Verma et al., 2019), replaces the original mini-batch data and gives rise to predicted probabilities $\widetilde{P} = (p_1, \ldots, p_m) \in \mathbb{R}^{c \times m}$ over classes, *e.g.* $\widetilde{P} = f(\widetilde{X})$ (Zhang et al., 2018) or $\widetilde{P} = g_W(\widetilde{Z})$ (Verma et al., 2019). Then, the average cross-entropy $H(\widetilde{Y}, \widetilde{P})$ (1) between the predicted probabilities $\widetilde{P}$ and interpolated targets $\widetilde{Y}$ is minimized. The number of interpolated data is $m$, same as the original mini-batch data.

## 3.2 MULTIMIX

**Interpolation** Given a mini-batch of $m$ examples with embeddings $Z$ and targets $Y$, we draw interpolation vectors $\lambda_k \sim \mathrm{Dir}(\alpha)$ for $k = 1, \ldots, n$, where $\mathrm{Dir}(\alpha)$ is the symmetric Dirichlet distribution and $\lambda_k \in \Delta^{m-1}$, that is, $\lambda_k \geq 0$ and $\mathbf{1}_m^\top \lambda_k = 1$. We then interpolate embeddings and targets by taking $n$ convex combinations over all $m$ examples:

$$\widetilde{Z} = Z\Lambda \tag{6}$$

$$\widetilde{Y} = Y\Lambda, \tag{7}$$

where $\Lambda = (\lambda_1, \ldots, \lambda_n) \in \mathbb{R}^{m \times n}$. We thus generalize manifold mixup (Verma et al., 2019):

1. from pairs to tuples of length $m$, as long as the mini-batch: $m$-term convex combination (6),(7) *vs.* 2-term in (3),(4), Dirichlet *vs.* Beta distribution;

2. from $m$ to an arbitrary number $n$ of tuples: interpolated embeddings $\widetilde{Z} \in \mathbb{R}^{d \times n}$ (6) *vs.* $\mathbb{R}^{d \times m}$ in (3), interpolated targets $\widetilde{Y} \in \mathbb{R}^{c \times n}$ (7) *vs.* $\mathbb{R}^{c \times m}$ in (4);

3. from fixed $\lambda$ across the mini-batch to a different $\lambda_k$ for each interpolated item.

**Loss** Again, we replace the original mini-batch embeddings $Z$ by the interpolated embeddings $\widetilde{Z}$ and minimize the average cross-entropy $H(\widetilde{Y}, \widetilde{P})$ (1) between the predicted probabilities $\widetilde{P} = g_W(\widetilde{Z})$ and the interpolated targets $\widetilde{Y}$ (7). Compared with (2), the mini-batch loss becomes

$$L_M(X, Y; \theta, W) := H(Y\Lambda, g_W(f_\theta(X)\Lambda)). \tag{8}$$

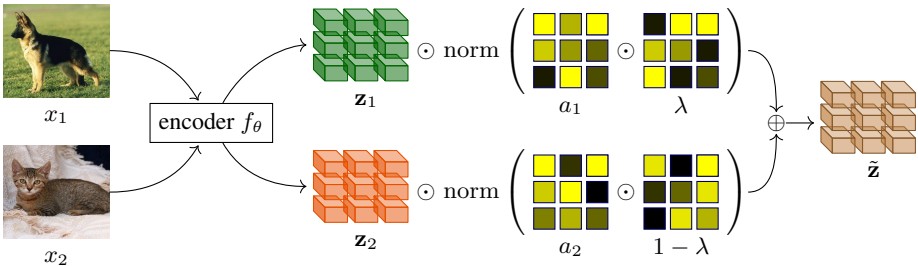

Figure 2: *Dense MultiMix* (subsection 3.4) for the special case $m = 2$ (two examples), $n = 1$ (one interpolated embedding), $r = 9$ (spatial resolution $3 \times 3$). The embeddings $\mathbf{z}_1, \mathbf{z}_2 \in \mathbb{R}^{d \times 9}$ of input images $x_1, x_2$ are extracted by encoder $f_\theta$. Attention maps $a_1, a_2 \in \mathbb{R}^9$ are extracted (10), multiplied element-wise with interpolation vectors $\lambda, (1 - \lambda) \in \mathbb{R}^9$ (11) and $\ell_1$-normalized per spatial position (12). The resulting weights are used to form the interpolated embedding $\tilde{\mathbf{z}} \in \mathbb{R}^{d \times 9}$ as a convex combination of $\mathbf{z}_1, \mathbf{z}_2$ per spatial position (13). Targets are interpolated similarly (14).

### 3.3 MULTIMIX WITH SELF-DISTILLATION

**Networks** We use an online self-distillation approach whereby the learned network $f := g_W \circ f_\theta$ becomes the *student*, whereas a *teacher* network $f' := g_{W'} \circ f_{\theta'}$ of the same architecture is obtained by exponential moving average of the parameters (Tarvainen & Valpola, 2017; Grill et al., 2020). The teacher parameters $(\theta', W')$ are not learned: We stop the gradient in the computation graph.

**Views** Given two transformations $T$ and $T'$, we generate two different augmented views $v = t(x)$ and $v' = t'(x)$ for each input $x$, where $t \sim T$ and $t' \sim T'$. Then, given a mini-batch of $m$ examples with inputs $X$ and targets $Y$, let $V = t(X), V' = t'(X) \in \mathbb{R}^{D \times m}$ be the mini-batch views corresponding to the two augmentations and $Z = f_\theta(V), Z' = f_{\theta'}(V') \in \mathbb{R}^{d \times m}$ the embeddings obtained by the student and teacher encoders respectively.

**Interpolation** We obtain the interpolated embeddings $\widetilde{Z}, \widetilde{Z}'$ from $Z, Z'$ by (6) and targets $\widetilde{Y}$ from $Y$ by (7), using the same $\Lambda$. The predicted class probabilities are given by $\widetilde{P} = g_W(\widetilde{Z})$ and $\widetilde{P}' = g_{W'}(\widetilde{Z}')$, again obtained by the student and teacher classifiers, respectively.

**Loss** We learn parameters $(\theta, W)$ by minimizing a classification and a self-distillation loss:

$$\gamma H(\widetilde{Y}, \widetilde{P}) + (1 - \gamma) H(\widetilde{P}', \widetilde{P}), \tag{9}$$

where $\gamma \in [0, 1]$. The former brings the probabilities $\widetilde{P}$ predicted by the student close to the targets $\widetilde{Y}$, as in (8). The latter brings $\widetilde{P}$ close to the probabilities $\widetilde{P}'$ predicted by the teacher.

### 3.4 DENSE MULTIMIX

We now extend to the case where the embeddings are structured, *e.g.* in tensors. This happens *e.g.* with token *vs.* sentence embeddings in NLP and patch *vs.* image embeddings in vision. This works by removing spatial pooling and applying the loss function densely over all tokens/patches. The idea is illustrated in Figure 2. For the sake of exposition, our formulation uses sets of matrices grouped either by example or by spatial position. In practice, all operations are on tensors.

**Preliminaries** The encoder is now $f_\theta : \mathcal{X} \to \mathbb{R}^{d \times r}$, mapping the input $x$ to an embedding $\mathbf{z} = f_\theta(x) \in \mathbb{R}^{d \times r}$, where $d$ is the number of channels and $r$ is its spatial resolution—if there are more than one spatial dimensions, these are flattened.

Given a mini-batch of $m$ examples, we have again inputs $X = (x_1, \ldots, x_m) \in \mathbb{R}^{D \times m}$ and targets $Y = (y_1, \ldots, y_m) \in \mathbb{R}^{c \times m}$. Each embedding $\mathbf{z}_i = f_\theta(x_i) = (z_i^1, \ldots, z_i^r) \in \mathbb{R}^{d \times r}$ for $i = 1, \ldots, m$ consists of features $z_i^j \in \mathbb{R}^d$ for spatial position $j = 1, \ldots, r$. We group features by position in matrices $Z^1, \ldots, Z^r$, where $Z^j = (z_1^j, \ldots, z_m^j) \in \mathbb{R}^{d \times m}$ for $j = 1, \ldots, r$.

**Attention** Each feature vector will inherit the target of the corresponding input example. However, we also attach a level of confidence according to an attention map. Given an embedding $\mathbf{z} \in \mathbb{R}^{d \times r}$ with target $y \in \mathcal{Y}$ and a vector $u \in \mathbb{R}^d$, the attention map

$$a = h(\mathbf{z}^\top u) \in \mathbb{R}^r \tag{10}$$

measures the similarity of features of $\mathbf{z}$ to $u$, where $h$ is a non-linearity, *e.g.* softmax or ReLU followed by $\ell_1$ normalization. There are different ways to define vector $u$. For example, $u = \mathbf{z}\mathbf{1}_r/r$ by global average pooling (GAP) of $\mathbf{z}$, or $u = Wy$ assuming a linear classifier with $W \in \mathbb{R}^{d \times c}$, similar to class activation mapping (CAM) (Zhou et al., 2016). In the case of no attention, $a = \mathbf{1}_r/r$ is uniform.

Given a mini-batch, let $a_i = (a_i^1, \ldots, a_i^r) \in \mathbb{R}^r$ be the attention map of embedding $\mathbf{z}_i$ (10). We group attention by position in vectors $a^1, \ldots, a^r$, where $a^j = (a_1^j, \ldots, a_m^j) \in \mathbb{R}^m$ for $j = 1, \ldots, r$.

**Interpolation** For each spatial position $j = 1, \ldots, r$, we draw interpolation vectors $\lambda_k^j \sim \text{Dir}(\alpha)$ for $k = 1, \ldots, n$ and define $\Lambda^j = (\lambda_1^j, \ldots, \lambda_n^j) \in \mathbb{R}^{m \times n}$. Because input examples are assumed to contribute according to the attention vector $a^j \in \mathbb{R}^m$, we scale the rows of $\Lambda^j$ accordingly and then we normalize its columns back to $\Delta^{m-1}$ so that they can define convex combinations:

$$M^j = \text{diag}(a^j)\Lambda^j \tag{11}$$

$$\hat{M}^j = M^j \text{diag}(\mathbf{1}_m^\top M^j)^{-1} \tag{12}$$

We then interpolate embeddings and targets by taking $n$ convex combinations over $m$ examples:

$$\widetilde{Z}^j = Z^j \hat{M}^j \tag{13}$$

$$\widetilde{Y}^j = Y \hat{M}^j. \tag{14}$$

This is similar to (6),(7), but there is a different interpolated embedding matrix $\widetilde{Z}^j \in \mathbb{R}^{d \times n}$ as well as target matrix $\widetilde{Y}^j \in \mathbb{R}^{c \times n}$ per position, even though the original target matrix $Y$ is one.

**Classifier** The classifier is now $g_W : \mathbb{R}^{d \times r} \to \mathbb{R}^{c \times r}$, maintaining the same spatial resolution as the embedding and generating one vector of predicted probabilities per spatial position. This is done by removing average pooling or any down-sampling operation. The interpolated embeddings $\widetilde{Z}^1, \ldots, \widetilde{Z}^r$ (13) are grouped by example into $\widetilde{\mathbf{z}}_1, \ldots, \widetilde{\mathbf{z}}_n \in \mathbb{R}^{d \times r}$, mapped by $g_W$ to predicted probabilities $\widetilde{\mathbf{p}}_1, \ldots, \widetilde{\mathbf{p}}_n \in \mathbb{R}^{c \times r}$ and grouped again by position into $\widetilde{P}^1, \ldots, \widetilde{P}^r \in \mathbb{R}^{c \times n}$.

In the simple case where the original classifier is linear, *i.e.* $W \in \mathbb{R}^{d \times c}$, it is seen as $1 \times 1$ convolution and applied densely to each column (feature) of $\widetilde{Z}^j$ for $j = 1, \ldots, r$.

**Loss** Finally, we learn parameters $\theta, W$ by minimizing the weighted cross-entropy $H(\widetilde{Y}^j, \widetilde{P}^j; s)$ of $\widetilde{P}^j$ relative to the interpolated targets $\widetilde{Y}^j$ again densely at each position $j$, where

$$H(Y, P; s) := -\mathbf{1}_c^\top (Y \odot \log(P))s/(\mathbf{1}_n^\top s) \tag{15}$$

generalizes (1) and the weight vector is defined as $s = \mathbf{1}_m^\top M^j \in \mathbb{R}^n$. This is exactly the vector used to normalize the columns of $M^j$ in (12). The motivation is that the columns of $M^j$ are the original interpolation vectors weighted by attention: A small $\ell_1$ norm indicates that for the given position $j$, we are sampling from examples of low attention, hence the loss is to be discounted.

## 4 EXPERIMENTS

### 4.1 SETUP

We use a mini-batch of size $m = 128$ examples in all experiments. For every mini-batch, we apply MultiMix with probability $0.5$ or input mixup otherwise. For MultiMix, the default settings are given in subsection 4.4. We follow the experimental settings of AlignMixup (Venkataramanan et al., 2022) and use PreActResnet-18 (R-18) (He et al., 2016) and WRN16-8 (Zagoruyko & Komodakis,

| DATASET | CIFAR-10 | | CIFAR-100 | | TI |
| NETWORK | R-18 | W16-8 | R-18 | W16-8 | R-18 |
|---|---|---|---|---|---|
| Baseline[†] | 95.41±0.02 | 94.93±0.06 | 76.69±0.26 | 78.80±0.55 | 56.49±0.21 |
| Manifold mixup (Verma et al., 2019)[†] | 97.00±0.05 | 96.44±0.02 | 80.00±0.34 | 80.77±0.26 | 59.31±0.49 |
| PuzzleMix (Kim et al., 2020)[†] | 97.04±0.04 | 97.00±0.03 | 79.98±0.05 | 80.78±0.23 | 63.52±0.42 |
| Co-Mixup (Kim et al., 2021)[†] | 97.10±0.03 | 96.44±0.08 | 80.28±0.13 | 80.39±0.34 | 64.12±0.43 |
| AlignMixup (Venkataramanan et al., 2022)[†] | 97.06±0.04 | 96.91±0.01 | 81.71±0.07 | 81.24±0.02 | 66.85±0.07 |
| $\zeta$-Mixup (Abhishek et al., 2022)[★] | 96.26±0.04 | 96.35±0.04 | 80.46±0.26 | 79.73±0.15 | 63.18±0.14 |
| MultiMix (ours) | 97.07±0.03 | 97.06±0.02 | 81.82±0.04 | 81.44±0.03 | 67.11±0.04 |
| + distil | 97.12±0.02 | 97.12±0.03 | 82.18±0.11 | 82.06±0.07 | 68.06±0.03 |
| + dense | 97.09±0.02 | 97.09±0.02 | 81.93±0.04 | 81.77±0.03 | 68.44±0.05 |
| + dense + distil | **97.16**±0.02 | **97.20**±0.02 | **82.35**±0.13 | **82.32**±0.03 | **69.11**±0.05 |
| Gain | **+0.06** | **+0.20** | **+0.64** | **+1.08** | **+2.26** |

Table 1: *Image classification* on CIFAR-10/100 and TI (TinyImagenet). Mean and standard deviation of Top-1 accuracy (%) for 5 runs. R: PreActResnet, W: WRN. ★: reproduced, †: reported by AlignMixup, **Bold black**: best; Blue: second best; underline: best baseline. Gain: improvement over best baseline. Comparison with additional baselines is given in subsection A.2

| NETWORK | RESNET-50 | | VIT-S/16 | |
| METHOD | SPEED | ERROR | SPEED | ERROR |
|---|---|---|---|---|
| Baseline[†] | 1.17 | 76.32 | 1.01 | 73.9 |
| Manifold mixup (Verma et al., 2019)[†] | 1.15 | 77.50 | 0.97 | 75.2 |
| PuzzleMix (Kim et al., 2020)[†] | 0.84 | 78.76 | 0.73 | 75.7 |
| Co-Mixup (Kim et al., 2021)[†] | 0.62 | – | 0.57 | 75.9 |
| AlignMixup (Venkataramanan et al., 2022)[†] | 1.03 | 79.32 | – | – |
| MultiMix (ours) | 1.16 | 78.81 | 0.98 | 75.2 |
| + distil | 1.06 | 80.12 | 0.93 | 76.7 |
| + dense | 0.95 | 79.32 | 0.88 | 76.1 |
| + dense + distil | 0.83 | **80.21** | 0.81 | **76.9** |
| Gain | | **+0.89** | | **+1.0** |

Table 2: *Image classification and training speed* on ImageNet. Top-1 accuracy (%): higher is better. Speed: images/sec ($\times 10^3$): higher is better. †: reported by AlignMixup. **Bold black**: best; Blue: second best; underline: best baseline. Gain: improvement over best baseline. Comparison with additional baselines is given in subsection A.2

2016b) as encoder on CIFAR-10 and CIFAR-100 datasets Krizhevsky et al. (2009); R-18 on Tiny-Imagenet (Yao & Miller, 2015) (TI); and Resnet-50 (R-50) and ViT-S/16 (Dosovitskiy et al., 2020) on ImageNet (Russakovsky et al., 2015).

We report the mean and standard deviation of the top-1 accuracy (%) for five runs on image classification. On robustness to adversarial attacks (subsection 4.2), we report the top-1 error. We also experiment on object detection (subsection 4.3) and out-of-distribution detection (subsection A.3).

## 4.2 RESULTS: IMAGE CLASSIFICATION AND ROBUSTNESS

**Image classification** In Table 1 we observe that MultiMix and Dense MultiMix already outperform SoTA on all datasets except CIFAR-10 with R-18, where they are on par with Co-Mixup. The addition of distillation increases the gain and outperforms SoTA on all datasets. Both distillation and dense improve over vanilla MultiMix and their effect is complementary on all datasets. On TI for example, distillation improves by 0.95%, dense by 1.33% and their combination by 2.0%. This combination brings an impressive gain of 2.26% over the previous SoTA – AlignMixup. We provide additional analysis of the embedding space on 10 classes of CIFAR-100 in subsection A.4.

In Table 2 we observe that on ImageNet with R-50, vanilla MultiMix outperforms all methods except AlignMixup. Adding dense, distillation or both outperforms all SoTA with both R-50 and ViT-S/16. Importantly, it brings an overall gain of 4% over the baseline with R-50 and 3% with ViT-S/16.

**Training speed** Table 2 shows the training speed of MultiMix and its variants compared with SoTA mixup methods, measured on NVIDIA V-100 GPU, including forward and backward pass. In

| ATTACK | FGSM | | | | | PGD | | | |
|---|---|---|---|---|---|---|---|---|---|
| DATASET | CIFAR-10 | | CIFAR-100 | | TI | CIFAR-10 | | CIFAR-100 | |
| NETWORK | R-18 | W16-8 | R-18 | W16-8 | R-18 | R-18 | W16-8 | R-18 | W16-8 |
| Baseline[†] | 88.8±0.11 | 88.3±0.33 | 87.2±0.10 | 72.6±0.22 | 91.9±0.06 | 99.9±0.0 | 99.9±0.01 | 99.9±0.01 | 99.9±0.01 |
| Manifold mixup (Verma et al., 2019)[†] | 76.9±0.14 | 76.0±0.04 | 80.2±0.06 | 56.3±0.10 | 89.3±0.06 | 97.2±0.01 | 98.4±0.03 | 99.6±0.01 | 98.4±0.03 |
| PuzzleMix (Kim et al., 2020)[†] | 57.4±0.22 | 60.7±0.02 | 78.8±0.09 | 57.8±0.03 | 83.8±0.05 | 97.7±0.01 | 97.0±0.01 | 96.4±0.02 | 95.2±0.03 |
| Co-Mixup (Kim et al., 2021)[†] | 60.1±0.05 | 58.8±0.10 | 77.5±0.02 | 56.5±0.04 | – | 97.5±0.02 | 96.1±0.03 | 95.3±0.03 | 94.2±0.01 |
| AlignMixup (Venkataramanan et al., 2022)[†] | 54.8±0.03 | 56.0±0.05 | 74.1±0.04 | 55.0±0.03 | 78.8±0.03 | 95.3±0.04 | 96.7±0.03 | 90.4±0.01 | 92.1±0.03 |
| $\zeta$-Mixup (Abhishek et al., 2022)[★] | 72.8±0.23 | 67.3±0.24 | 75.3±0.21 | 68.0±0.21 | 84.7±0.18 | 98.0±0.06 | 98.6±0.03 | 97.4±0.10 | 96.1±0.10 |
| MultiMix (ours) | 54.1±0.09 | 55.3±0.04 | 75.8±0.04 | 54.5±0.01 | 77.5±0.01 | 94.2±0.04 | 94.8±0.01 | 90.0±0.01 | 91.6±0.01 |
| + distillation | 52.5±0.05 | 51.4±0.01 | 73.5±0.03 | 52.7±0.02 | 76.2±0.05 | 92.6±0.01 | 93.9±0.02 | 88.8±0.01 | 90.5±0.01 |
| + dense | 54.1±0.01 | 53.3±0.03 | 74.5±0.03 | 52.9±0.04 | 75.5±0.04 | 92.9±0.04 | 92.6±0.01 | 88.6±0.03 | 90.8±0.01 |
| + dense + distillation | **52.0±0.03** | **50.1±0.04** | **73.0±0.02** | **52.1±0.02** | **75.1±0.01** | **90.8±0.01** | **90.5±0.03** | **87.5±0.01** | **90.1±0.03** |
| Gain | +2.8 | +5.9 | +1.1 | +2.9 | +3.7 | +4.5 | +5.6 | +2.9 | +2.0 |

Table 3: *Robustness to FGSM & PGD attacks*. Mean and standard deviation of Top-1 error (%) for 5 runs: lower is better. ★: reproduced, †: reported by AlignMixup. **Bold black**: best; Blue: second best; underline: best baseline. Gain: reduction of error over best baseline. TI: TinyImagenet. R: PreActResnet, W: WRN. Comparison with additional baselines is given in subsection A.2

terms of training speed, the vanilla MultiMix is on par with the baseline, bringing a gain of 2.49%. The addition of distillation is on par with SoTA AlignMixup, bringing a gain of 0.80%. Adding both dense and distillation brings a gain of 0.89% over AlignMixup, while being 19.4% slower. The inference speed is the same for all methods.

**Robustness to adversarial attacks** We follow the experimental settings of Align-Mixup (Venkataramanan et al., 2022) and use $8/255$ $l_\infty$ $\epsilon$-ball for FGSM (Goodfellow et al., 2015) and $4/255$ $l_\infty$ $\epsilon$-ball with step size 2/255 for PGD (Madry et al., 2018) attack. In Table 3 we observe that vanilla MultiMix is already more robust than SoTA on all datasets and settings except FGSM on CIFAR-100 with R-18, where it is on par with AlignMixup. The addition of dense, distillation or both again increases the robustness and shows that their effect is complementary. The overall gain is more impressive than in classification error. For example, against the strong PGD attack on CIFAR-10 with W16-8, the SoTA Co-Mixup improves the baseline by 3.8% and our best result improves the baseline by 9.4%, which is more than double.

## 4.3 RESULTS: TRANSFER LEARNING TO OBJECT DETECTION

We evaluate the effect of mixup on the generalization ability of a pre-trained network to object detection as a downstream task. Following the settings of CutMix (Yun et al., 2019), we pre-train R-50 on ImageNet with MultiMix and its variants and use it as the backbone for SSD (Liu et al., 2016) with fine-tuning on Pascal VOC07+12 (Everingham et al., 2010) and Faster-RCNN (Ren et al., 2015) with fine-tuning on MS-COCO (Lin et al., 2014).

In Table 4, we observe that, while vanilla MultiMix is slightly worse than AlignMixup, dense and distillation bring improvements over the SoTA on both datasets and are still complementary. This is consistent with classification results. Compared with the baseline, our best setting brings a gain of 2.40% mAP on Pascal VOC07+12 and 3.14% on MS-COCO.

| DATASET | VOC07+12 | MS-COCO |
|---|---|---|
| DETECTOR | SSD | FASTER R-CNN |
| Baseline[†] | 76.7 | 33.27 |
| Input mixup[†] | 76.6 | 34.18 |
| CutMix[†] | 77.6 | 35.16 |
| AlignMixup[†] | 78.4 | 35.84 |
| MultiMix (ours) | 77.9 | 35.73 |
| + distil | 78.7 | 35.97 |
| + dense | 78.5 | 35.89 |
| + dense + distil | **79.1** | **36.41** |
| Gain | +0.7 | +0.57 |

Table 4: *Transfer learning* to object detection. Mean average precision (mAP, %): higher is better. †: reported by AlignMixup. **Bold black**: best; Blue: second best; underline: best baseline. Gain: increase in mAP.

## 4.4 ABLATIONS

All ablations are performed using R-18 on CIFAR-100. For MultiMix, we study the effect of the layer where we interpolate, the number of tuples $n$ and a fixed value of Dirichlet parameter $\alpha$. More ablations are given in the supplementary material.

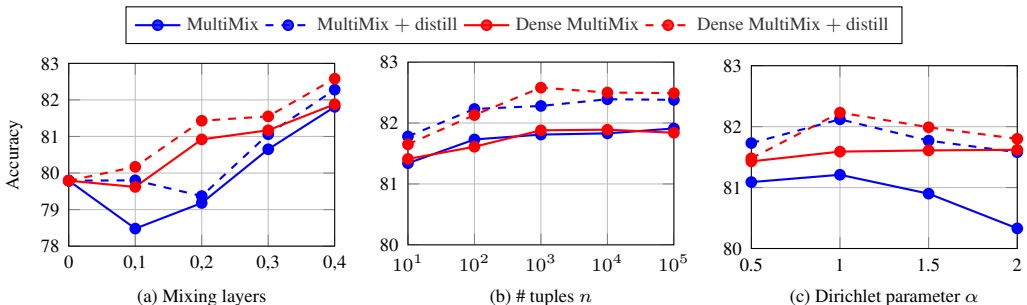

Figure 3: *Ablation study* of MultiMix and its variants on CIFAR-100 using R-18. (a) Interpolation layers (R-18 block; 0: input mixup). (b) Number of tuples $n$. (c) Dirichlet parameter $\alpha$.

**Interpolation layer**    For MultiMix, we use the entire network as the encoder $f_\theta$ by default, except for the last fully-connected layer, which we use as classifier $g_W$. Thus, we interpolate embeddings in the deepest layer by default. Here, we study the effect of different decompositions of the network $f = g_W \circ f_\theta$, such that interpolation takes place at a different layer. When using distillation, we interpolate at the same layer for both the teacher and the student. In Figure 3(a), we observe that mixing at the deeper layers of the network significantly improves performance. The same behavior is observed when adding dense, distillation, or both. This validates our default choice.

It is interesting that the authors of input mixup (Zhang et al., 2018) found that convex combinations of three or more examples in the input space with weights from the Dirichlet distribution do not bring further gain. This agrees with the finding of SuperMix (Dabouei et al., 2021) for four or more examples. Figure 3(a) suggests that further gain emerges when mixing in deeper layers.

**Number of tuples $n$**    Since our aim is to increase the amount of data seen by the model, or at least part of the model, it is important to study the number $n$ of interpolated embeddings. We observe from Figure 3(b) that accuracy increases overall with $n$ and saturates for $n \geq 1000$ for all variants of MultiMix. Our best setting, Dense MultiMix with distillation, works best at $n = 1000$. We choose this as default, given also that the training cost increases with $n$. The training speed as a function of $n$ is given in the supplementary material and is nearly constant for $n \leq 1000$.

**Dirichlet parameter $\alpha$**    Our default setting is to draw $\alpha$ uniformly at random from $[0.5, 2]$ for every interpolation vector (column of $\Lambda$). Here we study the effect of a fixed value of $\alpha$. In Figure 3(c), we observe that the best accuracy comes with $\alpha = 1$ for most MultiMix variants, corresponding to the uniform distribution over the convex hull of the mini-batch embeddings. However, all measurements are lower than the default $\alpha \sim U[0.5, 2]$. For example, from Table 1 (CIFAR-100, R-18), dense MultiMix + distillation has accuracy 82.52, compared with 82.23 in Figure 3(c) for $\alpha = 1$.

## 5    CONCLUSION

In terms of input interpolation, the take-home message of this work is that, instead of devising smarter and more complex interpolation functions in the input space or the first layers of the representation, it is more beneficial to just perform linear interpolation in the very last layer where the cost is minimal, and then increase as much as possible the number of interpolated embeddings for mixup. This is more in line with the original motivation of mixup as a way to go beyond ERM. In terms of target interpolation, the take-home message is the opposite: instead of linear interpolation of original targets, find new synthetic targets for the interpolated embeddings with the help of the network itself, then interpolate them linearly. This idea fits nicely with self-distillation, which is popular in settings such as self-supervised representation learning and continual learning. Interestingly, self-distillation can be seen as yet another form of augmentation, but in the model space.

A natural extension of this work is to settings other than supervised classification. A limitation is that it is not straightforward to combine the sampling scheme of MultiMix with complex interpolation methods, unless they are fast to compute in the embedding space.

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

# A  APPENDIX

# A  MORE EXPERIMENTS

## A.1  MORE ON SETUP

**Settings and hyperparameters**  We train MultiMix and its variants with mixed examples only. We use a mini-batch of size $m = 128$ examples in all experiments. For every mini-batch, we apply MultiMix with probability $0.5$ or input mixup otherwise. For input mixup, we interpolate the standard $m$ pairs (5). For MultiMix, we use the entire network as the encoder $f_\theta$ by default, except for the last fully-connected layer, which we use as classifier $g_W$. We use $n = 1000$ tuples and draw a different $\alpha \sim U[0.5, 2.0]$ for each example from the Dirichlet distribution by default. For multi-GPU experiments, all training hyperparameters including $m$ and $n$ are per GPU.

For dense MultiMix, the spatial resolution is $4 \times 4$ ($r = 16$) on CIFAR-10/100 and $7 \times 7$ ($r = 49$) on Imagenet by default. We obtain the attention map by (10) using GAP for vector $u$ and ReLU followed by $\ell_1$ normalization as non-linearity $h$ by default. To predict class probabilities and compute the loss densely, we use the classifier $g_W$ as $1 \times 1$ convolution by default; when interpolating at earlier layers, we follow the process described in subsection 3.4. For distillation, both the teacher and student networks have the same architecture. By default, we use $\gamma = \frac{1}{2}$ in (9), that is, equal contribution of original labels and teacher predictions.

**CIFAR-10/100 training**  Following the experimental settings of AlignMixup (Venkataramanan et al., 2022), we train MultiMix and its variants using SGD for 2000 epochs using the same random seed as AlignMixup. We set the initial learning rate to $0.1$ and decay it by a factor of $0.1$ every 500 epochs. The momentum is set to $0.9$ and the weight decay to $0.0001$. We use a batch size $m = 128$ and train on a single NVIDIA RTX 2080 TI GPU for 10 hours.

**TinyImageNet training**  Following the experimental settings of PuzzleMix (Kim et al., 2020), we train MultiMix and its variants using SGD for 1200 epochs, using the same random seed as AlignMixup. We set the initial learning rate to $0.1$ and decay it by a factor of $0.1$ after 600 and 900 epochs. The momentum is set to $0.9$ and the weight decay to $0.0001$. We train on two NVIDIA RTX 2080 TI GPUs for 18 hours.

**ImageNet training**  Following the experimental settings of PuzzleMix (Kim et al., 2020), we train MultiMix and its variants using the same random seed as AlignMixup. We train R-50 using SGD with momentum $0.9$ and weight decay $0.0001$ and ViT-S/16 using AdamW with default parameters. The initial learning rate is set to $0.1$ and $0.01$, respectively. We decay the learning rate by $0.1$ at 100 and 200 epochs. We train on 32 NVIDIA V100 GPUs for 20 hours.

**Tasks and metrics**  We use top-1 error (%, lower is better) or top-1 accuracy (%, higher is better) as evaluation metric on *image classification* and *robustness to adversarial attacks* (subsection 4.2 and subsection A.2). Additional datasets and metrics are reported separately for *transfer learning to object detection* (subsection 4.3) and *out-of-distribution detection* (subsection A.3).

## A.2  MORE RESULTS: CLASSIFICATION AND ROBUSTNESS

Using the experimental settings of subsection A.1, we extend Table 1, Table 2 and Table 3 of subsection 4.2 by comparing MultiMix and its variants with additional mixup methods in Table 6 and Table 7. The additional methods are Input mixup (Zhang et al., 2018), Cutmix (Yun et al., 2019), SaliencyMix (Uddin et al., 2021), StyleMix (Hong et al., 2021), StyleCutMix (Hong et al., 2021), SuperMix (Dabouei et al., 2021) and $\zeta$-Mixup (Abhishek et al., 2022). We reproduce $\zeta$-Mixup and SuperMix using these settings. For Supermix, we use the official code[1], which first trains the teacher network using clean examples and then the student using mixed. For fair comparison, we use the same network as the teacher and student models.

---

[1]https://github.com/alldbi/SuperMix

In Table 6 and Table 7, we observe that MultiMix and its variants outperform all the additional mixup methods on image classification. Furthermore, they are more robust to FGSM and PGD attacks as compared to these additional methods. The remaining observations in subsection 4.2 are still valid.

| Dataset | Cifar-10 | | Cifar-100 | | TI |
| Network | R-18 | W16-8 | R-18 | W16-8 | R-18 |
| --- | --- | --- | --- | --- | --- |
| Baseline[†] | $95.41_{\pm0.02}$ | $94.93_{\pm0.06}$ | $76.69_{\pm0.26}$ | $78.80_{\pm0.55}$ | $56.49_{\pm0.21}$ |
| Input mixup (Zhang et al., 2018)[†] | $95.98_{\pm0.10}$ | $96.18_{\pm0.06}$ | $79.39_{\pm0.40}$ | $80.16_{\pm0.1}$ | $56.60_{\pm0.16}$ |
| CutMix (Yun et al., 2019)[†] | $96.79_{\pm0.04}$ | $96.48_{\pm0.04}$ | $80.56_{\pm0.09}$ | $80.25_{\pm0.41}$ | $56.87_{\pm0.39}$ |
| Manifold mixup (Verma et al., 2019)[†] | $97.00_{\pm0.05}$ | $96.44_{\pm0.02}$ | $80.00_{\pm0.34}$ | $80.77_{\pm0.26}$ | $59.31_{\pm0.49}$ |
| PuzzleMix (Kim et al., 2020)[†] | $97.04_{\pm0.04}$ | $97.00_{\pm0.03}$ | $79.98_{\pm0.05}$ | $80.78_{\pm0.23}$ | $63.52_{\pm0.42}$ |
| Co-Mixup (Kim et al., 2021)[†] | $\underline{97.10}_{\pm0.03}$ | $96.44_{\pm0.08}$ | $80.28_{\pm0.13}$ | $80.39_{\pm0.34}$ | $64.12_{\pm0.43}$ |
| SaliencyMix (Uddin et al., 2021)[†] | $96.94_{\pm0.05}$ | $96.27_{\pm0.05}$ | $80.36_{\pm0.56}$ | $80.29_{\pm0.05}$ | $66.14_{\pm0.51}$ |
| StyleMix (Hong et al., 2021)[†] | $96.25_{\pm0.04}$ | $96.27_{\pm0.04}$ | $80.01_{\pm0.79}$ | $79.77_{\pm0.17}$ | $63.88_{\pm0.27}$ |
| StyleCutMix (Hong et al., 2021)[†] | $96.94_{\pm0.05}$ | $96.95_{\pm0.04}$ | $80.67_{\pm0.07}$ | $80.79_{\pm0.04}$ | $66.55_{\pm0.13}$ |
| SuperMix (Dabouei et al., 2021)[*] | $96.03_{\pm0.05}$ | $96.13_{\pm0.05}$ | $79.07_{\pm0.26}$ | $79.42_{\pm0.05}$ | $64.43_{\pm0.39}$ |
| AlignMixup (Venkataramanan et al., 2022)[†] | $97.06_{\pm0.04}$ | $96.91_{\pm0.01}$ | $\underline{81.71}_{\pm0.07}$ | $\underline{81.24}_{\pm0.02}$ | $66.85_{\pm0.07}$ |
| ζ-Mixup (Abhishek et al., 2022)[★] | $96.26_{\pm0.04}$ | $96.35_{\pm0.04}$ | $80.46_{\pm0.26}$ | $79.73_{\pm0.15}$ | $63.18_{\pm0.14}$ |
| MultiMix (ours) | $97.07_{\pm0.03}$ | $97.06_{\pm0.02}$ | $81.82_{\pm0.04}$ | $81.44_{\pm0.03}$ | $67.11_{\pm0.04}$ |
| + distil | $97.12_{\pm0.02}$ | $\mathbf{97.19}_{\pm0.03}$ | $82.18_{\pm0.11}$ | $82.06_{\pm0.07}$ | $68.06_{\pm0.03}$ |
| + dense | $97.09_{\pm0.02}$ | $97.09_{\pm0.02}$ | $81.93_{\pm0.04}$ | $81.77_{\pm0.03}$ | $68.44_{\pm0.05}$ |
| + dense + distil | $\mathbf{97.16}_{\pm0.02}$ | $\mathbf{97.20}_{\pm0.02}$ | $\mathbf{82.35}_{\pm0.13}$ | $\mathbf{82.32}_{\pm0.03}$ | $\mathbf{69.11}_{\pm0.05}$ |
| Gain | **+0.06** | **+0.20** | **+0.64** | **+1.08** | **+2.26** |

Table 5: *Image classification* on CIFAR-10/100 and TI (TinyImagenet). Top-1 accuracy (%): higher is better. R: PreActResnet, W: WRN. [*]: reproduced, [†]: reported by AlignMixup, [*]: reproduced with same teacher and student model. **Bold black**: best; Blue: second best; underline: best baseline. Gain: improvement over best baseline.

| Network | Resnet-50 | | ViT-S/16 | |
| Method | Speed | Error | Speed | Error |
| --- | --- | --- | --- | --- |
| Baseline[†] | 1.17 | 76.32 | 1.01 | 73.9 |
| Input mixup (Zhang et al., 2018)[†] | 1.14 | 77.42 | 0.99 | 74.7 |
| CutMix (Yun et al., 2019)[†] | 1.16 | 78.60 | 0.99 | 74.4 |
| Manifold mixup (Verma et al., 2019)[†] | 1.15 | 77.50 | 0.97 | 75.2 |
| PuzzleMix (Kim et al., 2020)[†] | 0.84 | 78.76 | 0.73 | 75.7 |
| Co-Mixup (Kim et al., 2021)[†] | 0.62 | – | 0.57 | $\underline{75.9}$ |
| SaliencyMix (Uddin et al., 2021)[†] | 1.14 | 78.74 | 0.96 | 75.8 |
| StyleMix (Hong et al., 2021)[†] | 0.99 | 75.94 | 0.85 | 74.8 |
| StyleCutMix (Hong et al., 2021)[†] | 0.76 | 77.29 | 0.71 | 75.1 |
| SuperMix (Dabouei et al., 2021)[*] | 0.92 | 77.60 | – | – |
| AlignMixup (Venkataramanan et al., 2022)[†] | 1.03 | $\underline{79.32}$ | – | – |
| MultiMix (ours) | 1.16 | 78.81 | 1.0 | 75.2 |
| + distil | 1.06 | 80.12 | 0.93 | 76.7 |
| + dense | 0.95 | 79.32 | 0.88 | 76.1 |
| + dense + distil | 0.83 | **80.21** | 0.81 | **76.9** |
| Gain | | **+0.89** | | **+1.0** |

Table 6: *Image classification and training speed* on ImageNet. Top-1 accuracy (%): higher is better. Speed: images/sec ($\times10^3$): higher is better. [*]: reproduced with same teacher and student model, [†]: reported by AlignMixup. **Bold black**: best; Blue: second best; underline: best baseline. Gain: improvement over best baseline.

## A.3 More results: Out of distribution detection

This is a standard benchmark for evaluating over-confidence. Here, *in-distribution* (ID) are examples on which the network has been trained, and *out-of-distribution* (OOD) are examples drawn from any other distribution. Given a mixture of ID and OOD examples, the network should predict an ID example with high confidence and an OOD example with low confidence, *i.e.*, the confidence of the predicted class should be below a certain threshold.

Following AlignMixup (Venkataramanan et al., 2022), we compare MultiMix and its variants with SoTA methods trained using R-18 on CIFAR-100 as ID examples, while using LSUN (Yu et al.,

| ATTACK | FGSM | | | | | PGD | | | |
|---|---|---|---|---|---|---|---|---|---|
| DATASET | CIFAR-10 | | CIFAR-100 | | TI | CIFAR-10 | | CIFAR-100 | |
| NETWORK | R-18 | W16-8 | R-18 | W16-8 | R-18 | R-18 | W16-8 | R-18 | W16-8 |
| Baseline[†] | 88.8±0.11 | 88.3±0.33 | 87.2±0.10 | 72.6±0.22 | 91.9±0.06 | 99.9±0.0 | 99.9±0.01 | 99.9±0.01 | 99.9±0.01 |
| Input mixup (Zhang et al., 2018)[†] | 79.1±0.07 | 79.1±0.12 | 81.4±0.23 | 67.3±0.06 | 88.7±0.08 | 99.7±0.02 | 99.4±0.01 | 99.9±0.01 | 99.3±0.02 |
| CutMix (Yun et al., 2019)[†] | 77.3±0.06 | 78.3±0.05 | 86.9±0.06 | 60.2±0.04 | 88.6±0.03 | 99.8±0.03 | 98.1±0.02 | 98.6±0.01 | 97.9±0.01 |
| Manifold mixup (Verma et al., 2019)[†] | 76.9±0.14 | 76.0±0.04 | 80.2±0.06 | 56.3±0.10 | 89.3±0.06 | 97.2±0.01 | 98.4±0.03 | 99.6±0.01 | 98.4±0.03 |
| PuzzleMix (Kim et al., 2020)[†] | 57.4±0.22 | 60.7±0.02 | 78.8±0.09 | 57.8±0.03 | 83.8±0.05 | 97.7±0.01 | 97.0±0.01 | 96.4±0.02 | 95.2±0.03 |
| Co-Mixup (Kim et al., 2021)[†] | 60.1±0.05 | 58.8±0.10 | 77.5±0.02 | 56.5±0.04 | – | 97.5±0.02 | 96.1±0.03 | 95.3±0.03 | 94.2±0.01 |
| SaliencyMix (Uddin et al., 2021)[†] | 57.4±0.08 | 68.0±0.05 | 77.8±0.10 | 58.1±0.06 | 81.1±0.06 | 97.4±0.03 | 97.0±0.04 | 95.6±0.03 | 93.7±0.05 |
| StyleMix (Hong et al., 2021)[†] | 80.0±0.23 | 71.2±0.21 | 80.6±0.15 | 68.2±0.17 | 85.1±0.16 | 98.1±0.09 | 97.5±0.07 | 98.3±0.09 | 98.3±0.09 |
| StyleCutMix (Hong et al., 2021)[†] | 57.7±0.04 | 56.0±0.07 | 77.4±0.05 | 56.8±0.03 | 80.5±0.04 | 97.8±0.04 | 96.7±0.02 | 91.8±0.01 | 93.7±0.01 |
| SuperMix (Dabouei et al., 2021)[*] | 60.0±0.11 | 58.2±0.12 | 78.8±0.13 | 58.3±0.19 | 81.1±0.12 | 97.6±0.02 | 97.2±0.09 | 91.4±0.03 | 92.7±0.01 |
| AlignMixup (Venkataramanan et al., 2022)[†] | 54.8±0.03 | 56.0±0.05 | 74.1±0.04 | 55.0±0.03 | 78.8±0.03 | 95.3±0.04 | 96.7±0.03 | 90.4±0.01 | 92.1±0.03 |
| ζ-Mixup (Abhishek et al., 2022)[*] | 72.8±0.23 | 67.3±0.24 | 75.3±0.21 | 68.0±0.21 | 84.7±0.18 | 98.0±0.06 | 98.6±0.03 | 97.4±0.10 | 96.1±0.10 |
| MultiMix (ours) | 54.1±0.09 | 55.3±0.04 | 75.8±0.01 | 54.5±0.01 | 77.5±0.01 | 94.2±0.04 | 94.8±0.01 | 90.0±0.01 | 91.6±0.01 |
| + distillation | 52.5±0.05 | 51.4±0.01 | 73.5±0.03 | 52.7±0.02 | 76.2±0.05 | 92.6±0.01 | 93.9±0.02 | 88.8±0.01 | 90.5±0.01 |
| + dense | 54.1±0.01 | 53.3±0.03 | 74.5±0.03 | 52.9±0.04 | 75.5±0.04 | 92.9±0.04 | 92.6±0.01 | 88.6±0.03 | 90.8±0.01 |
| + dense + distillation | **52.0±0.03** | **50.1±0.04** | **73.0±0.02** | **52.1±0.02** | **75.1±0.01** | **90.8±0.01** | **90.5±0.03** | **87.5±0.01** | **90.1±0.03** |
| Gain | +2.8 | +5.9 | +1.1 | +2.9 | +3.7 | +4.5 | +5.6 | +2.9 | +2.0 |

Table 7: *Robustness to FGSM & PGD attacks*. Top-1 error (%): lower is better. [*]: reproduced, [†]: reported by AlignMixup. [*]: reproduced, same teacher and student model. **Bold black**: best; Blue: second best; underline: best baseline. Gain: reduction of error over best baseline. TI: TinyImagenet. R: PreActResnet, W: WRN.

| TASK | OUT-OF-DISTRIBUTION DETECTION | | | | | | | | | | | |
|---|---|---|---|---|---|---|---|---|---|---|---|---|
| DATASET | LSUN (CROP) | | | | iSUN | | | | TI (CROP) | | | |
| METRIC | DET ACC | AUROC | AUPR (ID) | AUPR (OOD) | DET ACC | AUROC | AUPR (ID) | AUPR (OOD) | DET ACC | AUROC | AUPR (ID) | AUPR (OOD) |
| Baseline[†] | 54.0 | 47.1 | 54.5 | 45.6 | 66.5 | 72.3 | 74.5 | 69.2 | 61.2 | 64.8 | 67.8 | 60.6 |
| Input mixup (Zhang et al., 2018)[†] | 57.5 | 59.3 | 61.4 | 55.2 | 59.6 | 63.0 | 60.2 | 63.4 | 58.7 | 62.8 | 63.0 | 62.1 |
| Cutmix (Yun et al., 2019)[†] | 63.8 | 63.1 | 61.9 | 63.4 | 67.0 | 76.3 | 81.0 | 77.7 | 70.4 | 84.3 | 87.1 | 80.6 |
| Manifold mixup (Verma et al., 2019)[†] | 58.9 | 60.3 | 57.8 | 59.5 | 64.7 | 73.1 | 80.7 | 76.0 | 67.4 | 69.9 | 69.3 | 70.5 |
| PuzzleMix (Kim et al., 2020)[†] | 64.3 | 69.1 | 80.6 | 73.7 | 73.9 | 77.2 | 79.3 | 71.1 | 71.8 | 76.2 | 78.2 | 81.9 |
| Co-Mixup (Kim et al., 2021)[†] | 70.4 | 75.6 | 82.3 | 70.3 | 68.6 | 80.1 | 82.5 | 75.4 | 71.5 | 84.8 | 86.1 | 80.5 |
| SaliencyMix (Uddin et al., 2021)[†] | 68.5 | 79.7 | 82.2 | 64.4 | 65.6 | 76.9 | 78.3 | 79.8 | 73.3 | 83.7 | 87.0 | 82.0 |
| StyleMix (Hong et al., 2021)[†] | 62.3 | 64.2 | 70.9 | 63.9 | 61.6 | 68.4 | 67.6 | 60.3 | 67.8 | 73.9 | 71.5 | 78.4 |
| StyleCutMix (Hong et al., 2021)[†] | 70.8 | 78.6 | 83.7 | 74.9 | 70.6 | 82.4 | 83.7 | 76.5 | 75.3 | 82.6 | 82.9 | 78.4 |
| SuperMix (Dabouei et al., 2021)[*] | 70.9 | 77.4 | 80.1 | 72.3 | 71.0 | 76.8 | 79.6 | 76.7 | 75.1 | 82.8 | 82.5 | 78.6 |
| AlignMixup (Venkataramanan et al., 2022)[†] | 74.2 | 79.9 | 84.1 | 75.1 | 72.8 | 83.2 | 84.1 | 80.3 | 77.2 | 85.0 | 87.8 | 85.0 |
| ζ-Mixup (Abhishek et al., 2022)[*] | 68.1 | 73.2 | 80.8 | 73.1 | 72.2 | 82.3 | 82.2 | 79.4 | 74.4 | 84.3 | 82.2 | 77.2 |
| MultiMix (ours) | 79.2 | 82.6 | 85.2 | 77.6 | 75.6 | 85.1 | 87.8 | 83.1 | 78.3 | 86.6 | 89.0 | 88.2 |
| + distillation | 80.3 | 84.4 | 86.3 | 76.4 | 79.0 | 85.6 | 88.2 | **84.9** | 80.7 | 87.8 | 89.9 | 88.2 |
| + dense | 80.8 | 84.3 | 85.9 | 78.0 | 76.8 | 85.4 | 88.0 | 84.6 | 81.4 | 89.0 | **90.8** | 88.0 |
| + dense + distillation | **81.0** | **84.9** | **86.4** | **78.2** | **79.2** | **86.0** | **88.5** | 84.8 | **81.9** | **89.3** | **90.3** | **88.3** |
| Gain | +6.8 | +5.0 | +2.3 | +3.1 | +5.3 | +2.8 | +4.4 | +4.6 | +4.7 | +4.3 | +3.0 | +3.3 |

Table 8: *Out-of-distribution detection* using R-18. Det Acc (detection accuracy), AuROC, AuPR (ID) and AuPR (OOD): higher is better. [*]: reproduced, [†]: reported by AlignMixup. [*]: reproduced, same teacher and student model. **Bold black**: best; Blue: second best; underline: best baseline. Gain: increase in performance. TI: TinyImagenet.

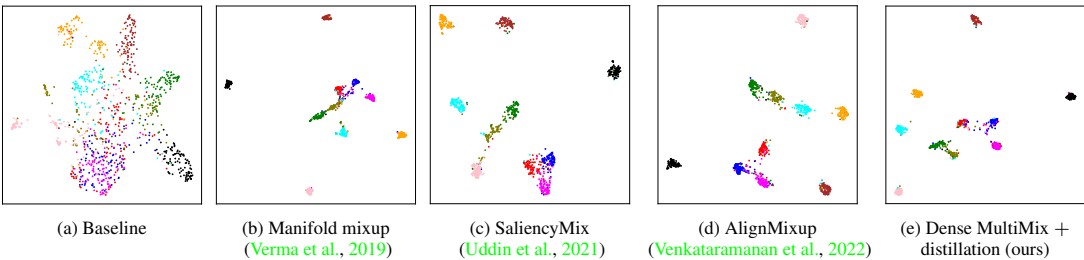

(a) Baseline    (b) Manifold mixup (Verma et al., 2019)    (c) SaliencyMix (Uddin et al., 2021)    (d) AlignMixup (Venkataramanan et al., 2022)    (e) Dense MultiMix + distillation (ours)

Figure 4: *Embedding space visualization* for 100 test examples per class of 10 randomly chosen classes of CIFAR-100 with PreActResnet-18, using UMAP (McInnes et al., 2018).

2015), iSUN (Xiao et al., 2010) and TI to draw OOD examples. We use detection accuracy, Area under ROC curve (AuROC) and Area under precision-recall curve (AuPR) as evaluation metrics. In Table 8, we observe that MultiMix and its variants outperform SoTA on all datasets and metrics by a large margin. Although the gain of vanilla MultiMix and Dense MultiMix over SoTA mixup methods is small on image classification, these variants significantly reduce over-confident incorrect predictions and achieve superior performance on out-of-distribution detection.

## A.4 ANALYSIS OF THE EMBEDDING SPACE

**Qualitative analysis** We qualitatively analyze the embedding space on 10 CIFAR-100 classes in Figure 4. We observe that the quality of embeddings of the baseline is extremely poor with severely overlapping classes, which explains its poor performance on image classification. All mixup methods result in clearly better clustered and more uniformly spread classes. Manifold mixup (Verma et al., 2019) produces five tightly clustered classes but the other five are still severely overlapping. SaliencyMix (Uddin et al., 2021) and AlignMixup (Venkataramanan et al., 2022) yield four somewhat clustered classes and 6 moderately overlapping ones. Our best setting, *i.e.*, dense MultiMix with distillation, results in five tightly clustered classes and another five somewhat overlapping but less than all competitors. More plots including variants of MultiMix are given in the supplementary material.

**Quantitative analysis** We also quantitatively assess the embedding space on the CIFAR-100 test set using alignment and uniformity (Wang & Isola, 2020). *Alignment* measures the expected pairwise distance of examples in the same class. Lower alignment indicates that the classes are more tightly clustered. *Uniformity* measures the (log of the) expected pairwise similarity of all examples using a Gaussian kernel as a similarity function. Lower uniformity indicates that classes are more uniformly spread in the embedding space. On CIFAR-100, we obtain alignment 3.02 for baseline, 1.27 for Manifold Mixup (Verma et al., 2019), 2.44 for SaliencyMix (Uddin et al., 2021), 2.04 for AlignMixup and 0.92 for Dense MultiMix with distillation. We also obtain uniformity -1.94 for the baseline, -2.38 for Manifold Mixup (Verma et al., 2019), -2.82 for SaliencyMix (Uddin et al., 2021), -4.77 for AlignMixup (Venkataramanan et al., 2022) and -5.68 for dense MultiMix with distillation. These results validate the qualitative analysis of Figure 4.

## A.5 MORE ABLATIONS

As in subsection 4.4, all ablations here are performed using R-18 on CIFAR-100.

**Mixup methods with distillation** In subsection 4.2 and Table 6, we observe that distillation significantly improves the performance when used with MultiMix. Here, we also study its effect when applied to SoTA mixup methods.

Given a mini-batch of $m$ examples with inputs $X$ and targets $Y$, we obtain the augmented views $V$ and $V'$ as discussed in subsection 3.3. We then follow the mixup strategy of each mixup method and obtain the corresponding predicted class probabilities $\widetilde{P}, \widetilde{P}'$ from the student and teacher classifier, respectively. *E.g.*, for manifold mixup (Verma et al., 2019), we interpolate the embeddings $Z = f_\theta(V), Z' = f_{\theta'}(V')$ using (3) and obtain $\widetilde{P} = g_W(\widetilde{Z})$ and $\widetilde{P}' = g_{W'}(\widetilde{Z}')$. In each case, we obtain the interpolated targets $\widetilde{Y}$ using (4) and train the student network using (9).

| METHOD | VANILLA | + DISTIL | + DENSE | + DENSE + DISTIL |
|---|---|---|---|---|
| Baseline | 76.76 | 78.28 | 78.16 | 79.07 |
| Input mixup (Zhang et al., 2018) | 79.79 | 80.19 | 80.21 | 80.54 |
| CutMix (Yun et al., 2019) | 80.63 | 81.51 | 81.40 | 81.61 |
| Manifold mixup (Verma et al., 2019) | 80.20 | 81.32 | 80.87 | 81.47 |
| PuzzleMix (Kim et al., 2020) | 79.99 | 81.26 | 80.62 | 81.44 |
| Co-Mixup (Kim et al., 2021) | 80.19 | 81.39 | 80.84 | 81.69 |
| SaliencyMix (Uddin et al., 2021) | 80.31 | 81.57 | 81.21 | 81.73 |
| StyleMix (Hong et al., 2021) | 79.96 | 81.22 | 80.76 | 81.30 |
| StyleCutMix (Hong et al., 2021) | 80.66 | 81.60 | 81.41 | 81.75 |
| SuperMix (Dabouei et al., 2021)* | 79.01 | 80.83 | 80.12 | 80.83 |
| AlignMixup (Venkataramanan et al., 2022) | 81.71 | 81.80 | 81.36 | 81.40 |
| MultiMix (ours)[‡] | – | – | 81.84 | 82.30 |
| MultiMix (ours) | **81.81** | **82.28** | **81.88** | **82.52** |

Table 9: *Image classification* on CIFAR-100 using R-18: The effect of distillation, dense loss and both on SoTA mixup methods. Top-1 accuracy (%): higher is better. *: 'vanilla' refers to teacher pre-training and 'distil' to self-distillation where teacher and student are trained concurrently from scratch. [‡]: Instead of dense MultiMix, we only apply the loss densely.

| METHOD | $u$ | $h$ | – | +DISTIL |
|---|---|---|---|---|
| Uniform | – | – | 81.33 | 81.59 |
| Attention (10) | CAM | softmax | 81.21 | 81.45 |
|  | CAM | $\ell_1 \circ$ relu | 81.63 | 81.91 |
|  | GAP | softmax | 81.78 | 82.01 |
|  | GAP | $\ell_1 \circ$ relu | **81.88** | **82.52** |

Table 10: *Variants of spatial attention* in dense MultiMix, with and without distillation, on CIFAR-100 using R-18. Top-1 accuracy (%): higher is better. GAP: Global Average Pooling; CAM: Class Activation Maps (Zhou et al., 2016); $\ell_1 \circ$ relu: ReLU followed by $\ell_1$ normalization.

In Table 9, we observe that with distillation, the performance of all SoTA mixup methods improve. For example, the baseline improves by 1.52% accuracy (76.76 → 78.23) and manifold mixup by 1.12% (80.20 → 81.32). On average, we observe a gain of 1% brought by distillation. An exception is AlignMixup (Venkataramanan et al., 2022): distillation brings a marginal improvement of 0.09% (81.71 → 81.80), making it on-par with vanilla MultiMix.

**Mixup methods with dense loss** In Table 6 we observe that dense interpolation and dense loss improve vanilla MultiMix. Here, we study the effect of the dense loss when applied to SoTA mixup methods.

Given a mini-batch of $m$ examples, we follow the mixup strategy of the SoTA mixup methods to obtain the mixed embedding $\widetilde{Z}^j \in \mathbb{R}^{d \times m}$ for each spatial position $j = 1, \ldots, r$. Then, as discussed in subsection 3.4, we obtain the predicted class probabilities $\widetilde{P}^j \in \mathbb{R}^{c \times m}$ again for each $j = 1, \ldots, r$. Finally, we compute the cross-entropy loss $H(\widetilde{Y}, \widetilde{P}^j)$ (1) densely at each spatial position $j$, where the interpolated target label $\widetilde{Y} \in \mathbb{R}^{c \times m}$ is given by (4).

In Table 9, we observe that using a dense loss improves the performance of all SoTA mixup methods. The baseline improves by 1.4% accuracy (76.76 → 78.16) and manifold mixup by 0.67% (80.20 → 80.87). On average, we observe a gain of 0.7% brought by the dense loss. An exception is AlignMixup (Venkataramanan et al., 2022), which drops by 0.35% (81.71 → 81.36). This may be due to the alignment process, whereby the interpolated dense embeddings are not very far from the original.

Finally, we study the effect of using a dense distillation loss on SoTA mixup methods. Here, similarly with (9), the loss has two terms for each spatial position $j$: the first is the dense cross-entropy loss $H(\widetilde{Y}, \widetilde{P}^j)$ as above and the second is the dense distillation loss $H((\widetilde{P}')^j, \widetilde{P}^j)$, where $\widetilde{P}'$ is obtained by the teacher. In Table 9, we observe that dense distillation further improves the performance of SoTA mixup methods as compared to using the dense loss only.

**Two-stage distillation**   Following SuperMix Dabouei et al. (2021), we also study the effect of using a two-stage distillation process with MultiMix, rather than online self-distillation.

In the first stage, we train the teacher using only clean examples for 300 epochs, and we achieve a top-1 accuracy of 75.62%. This is slightly lower than the 76.76% of the baseline from Table 9, which is trained for 2000 epochs. In the second stage, we fix the teacher parameters and train the student using the predictions from the teacher network as targets. In particular, we use the second term $H(\widetilde{P}', \widetilde{P})$ of (9), that is, $\gamma = 0$. At inference, the top-1 accuracy drops by 16% ($75.62 \rightarrow 59.77$). This shows that using the setting of SuperMix is not effective, while also being computationally expensive because of the two-stage training.

We also study the effect of training the student with both the interpolated labels $\widetilde{Y}$ (7) and the interpolated predictions $\widetilde{P}'$ of the pretrained teacher as targets. In particular, we use (9) with our default $\gamma = \frac{1}{2}$. At inference, the top-1 accuracy improves by 4.7% compared with the teacher ($75.62 \rightarrow 80.35$). However, the student accuracy of 80.35% is still inferior to our 82.28% by online self-distillation (Table 9). This shows that joint training of teacher and student is beneficial.

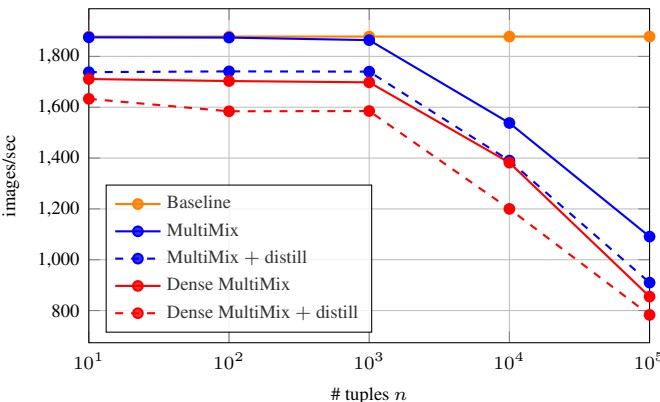

Figure 5: *Training speed* (images/sec) of MultiMix and its variants *vs.* number of tuples $n$ on CIFAR-100 using R-18. Measured on NVIDIA RTX 2080 TI GPU, including forward and backward pass.

**Training speed**   In Figure 5, we analyze the training speed of MultiMix and its variants as a function of number of tuples $n$. In terms of speed, vanilla MultiMix is on par with the baseline up to $n = 1000$, while bringing an accuracy gain of 5%. The best performing variant—dense MultiMix with distillation—is only slower by 15.6% at $n = 1000$ as compared to the baseline, which is arguably worth given the impressive 5.8% accuracy gain. Further increasing beyond $n > 1000$ brings a drop in training speed, due to computing $\Lambda$ and then using it to interpolate (6),(7). Because $n > 1000$ also brings little performance benefit according to Figure 3(b), we set $n = 1000$ as default for all MultiMix variants.

**Dense MultiMix: Spatial attention**   In subsection 3.4, we discuss different options for attention in dense MultiMix. In particular, no attention amounts to defining a uniform $a = \mathbf{1}_r/r$. Otherwise, $a$ is defined by (10). The vector $u$ can be defined as $u = \mathbf{z}\mathbf{1}_r/r$ by global average pooling (GAP) of $\mathbf{z}$, which is the default, or $u = Wy$ assuming a linear classifier with $W \in \mathbb{R}^{d \times c}$. The latter is similar to class activation mapping (CAM) Zhou et al. (2016), but here the current value of $W$ is used online while training. The non-linearity $h$ can be softmax or ReLU followed by $\ell_1$ normalization ($\ell_1 \circ \mathrm{relu}$), which is the default. Here, we study the affect of these options on the performance of dense Multimix.

In Table 10, we observe that using GAP for $u$ and $\ell_1 \circ \mathrm{relu}$ as $h$ yields the best performance overall. Changing GAP to CAM or $\ell_1 \circ \mathrm{relu}$ to softmax is inferior, more so in the presence of distillation. The combination of CAM with softmax is the weakest, even weaker than uniform attention. CAM may fail because of using the non-optimal value of $W$ while training; softmax may fail because of being too selective. Compared to our best result, uniform attention is clearly inferior, by nearly 1% in the presence of distillation. This validates that the use of spatial attention in dense MultiMix is clearly beneficial. The intuition is the same as in weakly supervised tasks: In the absence of dense

targets, assuming the same target of the entire example at every spatial position naively implies that the object of interest is present everywhere, whereas spatial attention provides a better hint as to where the object may really be.

**Dense MultiMix: Spatial resolution**    We study the effect of spatial resolution on dense MultiMix. By default, we use a resolution of $4 \times 4$ at the last residual block of R-18 on CIFAR-100. Here, we additionally investigate $1 \times 1$ (downsampling by average pooling with kernel size 4, same as GAP), $2 \times 2$ (downsampling by average pooling with kernel size 2) and $8 \times 8$ (upsampling by using stride 1 in the last residual block). We measure accuracy 81.07% for spatial resolution $1 \times 1$, 81.43% for for $2 \times 2$, 81.88% for $4 \times 4$ and 80.83% for $8 \times 8$. We thus observe that performance improves with spatial resolution up to $4 \times 4$, which the optimal, and then drops at $8 \times 8$. This drop may be due to assuming the same target at each spatial position. The resolution $8 \times 8$ is also more expensive computationally.

