# OpenReview forum: "Teach me how to Interpolate a Myriad of Embeddings"
_ICLR.cc/2023/Conference — Submitted to ICLR 2023_

### Official Review · Reviewer_Vnzf · 2022-10-21

**Confidence:** 3
**Correctness:** 3
**Technical Novelty And Significance:** 2
**Empirical Novelty And Significance:** 2
**Recommendation:** 5

**Clarity, Quality, Novelty And Reproducibility:**

* Clarity: The paper is well-written.
* Quality:  The experiments are extensive, namely, they are conducted on various applications and datasets.
* Novelty: The idea of Multi-Mix is not very original. However, the extension to Dense Mult-Mix is new to the best of my knowledge.
* Reproducibility: The code is not submitted, however, experimental settings are reported carefully.

**Strength And Weaknesses:**

## Strength

* The paper is well-organized and easy to follow.
* The experiments are conducted on various datasets and applications. The proposed techniques improve the performance slightly.
* Multimix is more robust to the adversarial attack than the previous baselines.
* The dense Multimix idea is interesting to perform mixup techniques in deep learning.

## Weaknesses

* The idea of using more samples to do mixing is not new. The authors should include those variants in the comparison. The authors have mentioned $m$-Mixup. The other variant that the paper should include is $k$-MIXUP [1] which also considers $m$ samples as MultiMix, however, $k$-mixup finds the optimal mapping between two mini-batches of size $m$ via optimal transport.

[1] k-Mixup Regularization for Deep Learning via Optimal Transport, Kristjan Greenewald, et al

* MultMix needs to sacrifice the computation time for the performance. However, I see that the improvement in performance is not very significant while the computational time is only reported in the image classification application. The authors should report the computational time for all other applications for completeness.

**Summary Of The Paper:**

The paper proposes MultiMix which is a variant of the Mixup data augmentation technique. While Mixup creates augmented data by only considering interpolation between two data points, MultiMix considers the interpolation between $m$ data samples. In greater detail, MultiMix creates $n$ new augmented samples from $m$ data samples by linear mixing them with random weight vectors that are drawn from a Dirichlet distribution.  Moreover, the authors extend MultiMix to Dense-Multimix by removing spatial pooling and applying the loss function densely over all tokens/patches in some special deep-learning architectures. In the case that has label information, the author uses online self-distillation for creating soft labels for augmented data. On the application side, the authors conduct experiments in image classification, robustness to adversarial attacks, object detection, and out-of-distribution detection. The proposed techniques improve the classification accuracies on CIFAR10, CiFAR100, and TI better than baselines in both clean and adversarial attacked data. The proposed techniques also yield better accuracy in objective detection on VOC07+12 (SSD model) and MS-COCO (FASTER R-CNN model).

**Summary Of The Review:**

The paper misses some important baselines that also consider $m$ samples for creating new augmented data as mentioned in the weakness part. Moreover, the trade-off between performance and computation is not well-studied. A figure is needed with the x-axis as computational time and the y-axis as the performance.

---

> ### Author Response · Authors · 2022-11-14
> **Response to Reviewer Vnzf**
>
> We appreciate the Reviewer Vnzf's valuable feedback. We address the concerns as follows:
>
> **1. Idea of using more samples**
> We never claim that we are the first to interpolate more than two examples. We state that using manifold mixup as a starting point, we extend and generalize it in a number of ways, one of which is to interpolate tuples as large as mini-batch size. This implies that we interpolate in the embedding space. By contrast, other existing methods that interpolate more than two examples, e.g SuperMix, do so in the input space. The difference is crucial, because, as Reviewer rQBR acknowledges, "*previous papers show mixup more examples doesn't help, which is contrary to what this paper argues*''. In fact, we discuss this at the end of the "Interpolation layer'' paragraph in Section 4.4, in connection with Figure 3(a).
>
> **2. $k$-Mixup**
> We thank the reviewer for referring us to this paper. $k$-Mixup does not interpolate more than two examples but uses more pairs than the mini-batch size. In MultiMix, most of the gain comes from increasing the size of the tuples rather than the number of tuples---see point 4 of our response to Reviewer rQBR. Furthermore, $k$-Mixup interpolates in the input space, while MultiMix interpolates in the embedding space. Please refer to point 1 above for this difference. To the best of our knowledge, $k$-Mixup does not have open-source code nor has it been published in a peer-reviewed venue.
>
> **3. Accuracy vs throughput tradeoff**
> The reviewer claims that "*MultMix needs to sacrifice the computation time for the performance*''. This is not true. For example, in ImageNet-1k classification, Table 2, MultiMix$+$distil yields an accuracy of 80.12 at $1.06 \times 10^3$ images/sec, which is better and faster than the SoTA AlignMixup with an accuracy of 79.32 at $1.03 \times 10^3$ images/sec. Reviewer rQBR describes the situation more accurately: ``*the proposed model has better accuracy and higher efficiency*''.
>
> As for other applications, we state in Section 4.2 under "training speed" that "*the inference speed is the same for all methods*.'' It is well understood that mixup, and more generally data augmentation only affects training throughput/efficiency.
>
> The reviewer also ignores that, other than standard accuracy, the improvements brought in other properties, like robustness (Tables 3,7) and reduced overconfidence (Table 8), are much more pronounced, e.g. up to 5 or 6\%.
>
> An accuracy *vs.* throughput plot is an interesting idea. We shall include one in the Appendix.

---

> > ### Comment · Reviewer_Vnzf · 2022-11-15
> > **Response to the authors**
> >
> > Thank you for your response.
> >
> > For 1 and 2, thank you for clarifying the difference between your work and past works.
> >
> > For 3, could you include the training speed (not inference speed) of the proposed techniques and previous baselines in Table 3 and Table 4?

---

> > > ### Author Response · Authors · 2022-11-16
> > > **Follow-up response to Reviewer Vnzf**
> > >
> > > We thank the reviewer for acknowledging that we clarify the reviewer's initial two queries.
> > >
> > > 3. We apologize if this is not clear enough. We mention in Section 4.2 under “Robustness to adversarial attacks”, that we follow the experimental settings of AlignMixup. To clarify, we train our models (R-18 and W 16-8) on CIFAR-10/100 and TinyImagenet for image classification (Table 1) and, once these models are trained, we apply FGSM and PGD attacks at inference (Table 3). Thus the training time for Table 3, would refer to the training time in Table 1. Similar to all mixup methods, we do not explicitly train our models on adversarial examples.
> > >
> > > On CIFAR-100 for image classification using R-18, the throughput (images/sec $\times 10^3$) is given as
> > > |                 | Baseline | Manifold | PuzzleMix | Co-Mixup | AlignMixup| $\zeta$-Mixup | MultiMix | MultiMix$+$distil | MultiMix$+$dense | MultiMix$+$dense$+$distil |
> > > |------------|----------|----------|-----------|----------|------------|----------|----------|-----------|----------|----------|
> > > | Throughput (img/sec $\times 10^3$) | 1.72     | 1.70     | 1.18      | 0.87     | 1.54     | 1.68     | 1.71      | 1.62     |1.48     | 1.29     |
> > >
> > >
> > > We shall add the training time for all models, methods and datasets for Table 1 in the Appendix.
> > >
> > > For object detection, we use R-50 pretrained on ImageNet and fine-tune it as the backbone on Faster-RCNN and SSD. Mixup is only used at ImageNet pretraining, not at fine-tuning. Thus the training time for Table 4, would be the same as that for image classification on ImageNet-1k (Table 2).
> > >
> > > We thank the reviewer for the comments and kindly request the reviewer to raise the ratings if they think that we have indeed addressed their concerns.

---

> > > > ### Comment · Reviewer_Vnzf · 2022-11-17
> > > > **Response to the authors**
> > > >
> > > > Thank you for your response. All my questions have been addressed. I have raised my score to 5. Due to my lack of knowledge of the literature on mixups, I will adjust my score again based on other discussions.
> > > >
> > > > Best,

---

### Official Review · Reviewer_rQBR · 2022-10-24

**Confidence:** 4
**Correctness:** 3
**Technical Novelty And Significance:** 2
**Empirical Novelty And Significance:** 2
**Recommendation:** 3

**Clarity, Quality, Novelty And Reproducibility:**

* Clarity: the paper is overall well written
* quality: fair, contribution is somewhat incremental
* novelty: fair, it feels like combining multiple existing techniques together, in an orthogonal way
* reproducibility: seems good


**Strength And Weaknesses:**

strengths
* proposed an extension of mixup with better accuracy and higher efficient

Weakness
* the contribution seems a bit incremental
* it’s unclear how the dense mixup and self-distillation are integral part of the proposed method, can we apply the same techniques to the baseline methods?


**Summary Of The Paper:**

This paper proposes an extension of the mixup data augmentation method where all examples in a batch are mixed with coefficients sampled from Dirichlet distribution, and the number of mixup examples can be as many as one wishes with more Dirichlet samples. On top of this, the method further adds ensemble distillation and dense augmentation and losses, which is shown to further improve the accuracy.

The paper argues that mixup on the top layer before logit is the best and also most efficient.


**Summary Of The Review:**

This paper proposed an extension of the mixup data augmentation method combined with some other techniques such as dense augmentation/loss and self-distillation with ensemble prediction. Though it demonstrates slightly better accuracy, it’s unclear how the different parts (mixup, dense augmentation, self-distillation) interact with each other.

questions
* in experiments, half of the time it’s using put augmentation, the other half MultiMix, unclear how this hyperparameter affect performance. Do we must have input mixup? Why?
* What prevents us from applying dense augmentation and self-distillation to other mixup baselines?
* for manifold mixup, did you also apply mixup in the last layer?
* previous papers show mixup more examples doesn't help, which is contrary to what this paper argues. But there is no ablation study to still mixup two examples, but mixup more batches. How do we know the gain is from mixing up more examples or simply because we have more tuples?
* high-level question: penultimate layer is already linear representation (i.e., input of logistic regression), and mixup is intend to regularize by enforcing linearity of the representation, in this case, why would it still help to apply mixup? In another work, would it help to apply mixup for logistic regression? (or softmax for multi-lass)

---

> ### Author Response · Authors · 2022-11-14
> **Response to Reviewer rQBR**
>
> We appreciate the Reviewer rQBR's valuable feedback. We address the concerns as follows:
>
> **1. Using input mixup**
> In Section 4.1, we mention that our experimental settings follow AlignMixup, which in turn follows manifold mixup. According to these settings, the probability of choosing between Input mixup and Multimix is 0.5. CutMix also uses the same setting. It is important to note that most SoTA mixup methods like CutMix, Co-Mixup, StyleCutMix etc. interpolate in the input space, while very few methods operate in the feature space, e.g manifold mixup, AlignMixup and $\zeta$-Mixup. Since the 0.5 probability is common for these latter methods, it is also essential that we adopt it for fair comparisons. Finally, the 0.5 probability in choosing between different mixup strategies is also standard when training transformers, e.g in the timm library.
>
> The 0.5 probability works best for MultiMix, as for previous work. On CIFAR-100 for example, we empirically observe that by using a probability of 0.8 (80\% input mixup and 20\% MultiMix), the accuracy drops from 81.81\% (Table 9) $\rightarrow$ 77.39\%, while with a probability of 0.2, the accuracy drops from 81.81\% $\rightarrow$ 79.91\%.
>
> Still on CIFAR-100, when input mixup is not used (probability 0.0), the accuracy of manifold mixup drops from 80.20 (Table 9)$\rightarrow$79.09\% and the accuracy of AlignMixup drops from 81.71$\rightarrow$80.28\% (from Table 6 of AlignMixup). For MultiMix, we report here that the drop is from 81.81 (Table 9)$\rightarrow$80.59\%, which is comparable with AlignMixup and manifold mixup.
>
> We shall add an ablation on the input mixup probability in the Appendix.
>
> **2. Dense augmentation and distillation to other methods.**
> We apply dense loss and self-distillation to all mixup methods in Table 9 of the Appendix. Dense augmentation is not straightforward or not possible to apply to any given mixup method.
>
> **3. Manifold mixup in the last layer?**
> Yes, for fair comparisons with MultiMix, we reproduce manifold mixup by performing mixup in the last layer. We empirically observe from AlignMixup that interpolating in the last layer results in better performance than earlier layers. See point 5 below.
>
> **4. Mixing more examples in a tuple or more tuples**
> We thank the reviewer for this question. In MultiMix we interpolate $n$ tuples, each of length $m$, where $m$ is the mini-batch size. We provide an ablation of the number of tuples $n$ on CIFAR-100 in Figure 3(b). This Figure shows that for $n=100$ and $m=128$, we achieve top-1 accuracy of 81.73\% and as $n$ increases, the performance saturates. To understand the effect of the number of examples in a tuple, consider manifold mixup with $n=128$ and $m=2$, which results in a CIFAR-100 accuracy of 80.20\% (Table 9). Thus, interpolating more examples in a tuple while having roughly the same number of tuples results in a 1.5\% accuracy gain.
>
> We did not ablate the size of the tuples, because this would involve changing our entire method, \ie sampling a subset of $m$ examples from the mini-batch, then sampling from Dirichlet. This would make the method more complex and less elegant. However, we shall add manifold mixup ($m=2$) in Figures 3(a,b,c) to show the effect of mixing layer, the number of pairs $n$ and Beta parameter $\alpha$. This will clarify the effect of going from $m=2$ to the mini-batch size.
>
> **5.  How does mixing help regularize penultimate layer?**
> We thank the reviewer for this interesting question. Manifold mixup argues that, since high-level representations are often low-dimensional and useful to linear classifiers, linear interpolations of feature space explore meaningful regions effectively. Even though the remaining mapping is in just one linear layer in MultiMix, which looks trivial, we observe that interpolating in the deeper layers greatly improves the representation of the model. This is clearly seen in Figure 3(a). The same finding is also confirmed in AlignMixup. This points to another advantage of MultiMix, that linear interpolation of a large number of tuples can be possible at a *low cost*, only when mixup is performed in the penultimate layer.

---

> > ### Author Response · Authors · 2022-11-17
> > **Request for Reviewer rQBR's feedback**
> >
> > Dear Reviewer rQBR
> >
> > Thank you again for reading our rebuttal. We believe we have faithfully given a detailed response to your concerns. We would appreciate it if you could please let us know if our response has sufficiently addressed your questions and thus kindly reconsider your score. We are also willing to clarify in case there are any further queries.
> >
> > Thank you.

---

### Official Review · Reviewer_BWYW · 2022-10-26

**Confidence:** 4
**Clarity, Quality, Novelty And Reproducibility:** The paper is rigorously written and e…
**Correctness:** 3
**Technical Novelty And Significance:** 3
**Empirical Novelty And Significance:** 3
**Recommendation:** 6

**Strength And Weaknesses:**

Strength:
1. The notations in the paper are rigorous.
2. The writing of the paper is easy to follow.
3. The proposed techniques are empirically solid.

Weakness:
1. None of the proposed techniques is theoretically justified. In other words, it is hard to judge in advance whether an approach helps before actually trying it out. Therefore, describing some scenarios where these techniques FAIL may be helpful.
2. It looks to me that the paper is an ensemble of tricks for training neural networks --- It does not solve any problem.
3. The scale of the experiments is limited --- the largest dataset is Tiny ImageNet. While this scale may be sufficient for a theoretical paper, it is less convincing for an empirical paper.

**Summary Of The Paper:**

The paper proposes a suite of Mix-based augmentation techniques that improve the performance of neural networks.

**Summary Of The Review:**

Overall, I think the proposed augmentation techniques are empirically sound, which expands the toolbox of augmentations for training neural networks. However, none of these techniques has been theoretically justified; therefore, it is hard to judge whether these techniques are applicable in a given (new) scenario.

---

> ### Author Response · Authors · 2022-11-14
> **Response to Reviewer BWYW**
>
> We appreciate the Reviewer BWYW 's valuable feedback. We address the concerns as follows:
>
> **1. Theoretical justification and scenarios where MultiMix may fail.**
> The reviewer claims that *``it is hard to judge whether an approach helps before trying it out."* This applies to all mixup methods in general. In this sense, no such method is accompanied by theoretical guarantees on accuracy or other studied properties, like calibration or robustness. Hence, all such methods are supported by empirical evidence and MultiMix is no exception. We cannot see how this claim is a weakness of MultiMix.
>
> In addition, failure cases (e.g misclassified examples that are correctly classified in the absence of mixup) do not directly explain how mixup harms, similarly to any regularizer that may have a positive or negative impact on accuracy, on average. We are not aware of any mixup method that discusses failure cases.
>
> To better understand *why* MultiMix works beyond average metrics, we visualize the embedding space in Subsection A.4. In particular, we show that MutliMix results in tight clusters that are uniformly spread over the hypersphere. We also evaluate these properties quantitatively, using alignment and uniformity metrics.
>
> **2. ``An ensemble of tricks for training neural networks - It does not solve any problem**"
> Put correctly, MultiMix is an ensemble of ideas to improve neural network training by data augmentation, like all mixup methods. In this sense, it is not clear what the reviewer means by ``*It does not solve any problem''*. We kindly ask for a detailed account with supporting evidence of how exactly it is a limitation or weakness of MultiMix, in the particular context of mixup-based data augmentation. For example, how other mixup methods solve a problem where MultiMix does not.
>
> **3. Scale of experiments**
> We evaluate MultiMix on ImageNet-1K using Resnet-50 and ViT-S/16 in Table 2. The complete set of comparisons with SoTA mixup methods on ImageNet-1K is in table 6 in the Appendix.
>  It is strange that the reviewer missed this because we also evaluate MultiMix on MS-COCO object detection (Table 4), where we use as backbone a Resnet-50 pretrained on ImageNet-1K with different mixup methods including MultiMix. Excluding ablations, the results given in the main paper consist of four tables, two of which (Table 2 and Table 4) are on large-scale datasets.

---

### Author Response · Authors · 2022-11-15
**Common response to Reviewer BWYW, rQBR and Vnzf**

We thank all the reviewers for their time and valuable feedback. We are grateful that the reviewers find that

- The extension to Dense MultMix is *new and interesting* (R-Vnzf).
- The paper is *well-organized* and easy to follow (R-BWYW, R-rQBR and R-Vnzf).
- The proposed idea has *better accuracy* and *higher efficiency* (R-rQBR), is *empirically solid* (R-BWYW) and *more robust* to adversarial attacks (R-Vnzf); and that the experiments are *extensive* (R-Vnzf).

Reviewer rQBR suggests a few additional ablations, i.e. to study the effect of input mixup and the effect of mixing more examples in a tuple. We believe those are interesting and shall add them in the Appendix.
However, we are bothered that the remaining of the reviewers' concerns are mostly:
- **Asking for things already in the paper**: Large-scale experiments on ImageNet-1K for image classification and MS-COCO for object detection, already in Tables 2, 4 (R-BWYW); applying dense augmentation and self-distillation to all mixup methods, already in Table 9 (R-rQBR).
- **Incorrect claims**: We never claim we are the first to mix multiple examples; Multimix is not similar to $k$-Mixup; MultiMix does not sacrifice computation time for performance (R-Vnzf); Every mixup method is hard to judge that it helps before trying it out; Mixup acts as a regularizer and failure cases does not explain how it harms (R-BWYW).
- **Unclear comments**: How other mixup methods solve a problem where MultiMix does not (R-BWYW).

These concerns do not indicate any *major flaw* that justifies rejection or helps us improve this work. We address them in our responses to individual reviews. We kindly request the reviewers to either improve their scores in case the concerns are indeed addressed or provide more detailed comments with more supporting evidence that will justify the scores better and help us improve.

---

### Decision · Program_Chairs · 2023-01-20

**Decision:**

Reject

**Justification For Why Not Higher Score:**

The method is fairly straightforward, which on its own would not be a problem --- however, the performance gains are pretty marginal (and come at an expected marginal computational cost).

**Justification For Why Not Lower Score:**

N/A

**Metareview: Summary, Strengths And Weaknesses:**

The paper proposes MultiMix, a strategy to extend prior MixUp style techniques to allow mixing k samples (rather than just a pair). The idea is fairly straightforward: one generalizes the $(\lambda, 1-\lambda)$ convex weights to samples for mixing coefficients from a Dirichlet distribution (which is well-defined for any number of samples k). The authors test on several datasets (testing classification, transfer, robustness) and few ablations (in particular w and w/o distillation). Overall, the feeling of the reviewers was that while the method is well-described, and the paper is easy to read and re-implement the algorithm, the improvements gained on all the datasets considered are not particular big (especially since the training computation cost rises).